# Quality Characterization of Different Parts of Broiler and Ligor Hybrid Chickens

**DOI:** 10.3390/foods11131929

**Published:** 2022-06-28

**Authors:** Worawan Panpipat, Manat Chaijan, Supatra Karnjanapratum, Pensiri Keawtong, Pavit Tansakul, Atikorn Panya, Natthaporn Phonsatta, Kittipat Aoumtes, Tran Hong Quan, Tanyamon Petcharat

**Affiliations:** 1Food Technology and Innovation Research Center of Excellence, Department of Food Industry, School of Agricultural Technology and Food Industry, Walailak University, Nakhon Si Thammarat 80161, Thailand; pworawan@wu.ac.th (W.P.); cmanat@wu.ac.th (M.C.); supatra.ka@mail.wu.ac.th (S.K.); pensiri.ka@mail.wu.ac.th (P.K.); 2Professional Culinary Arts Program, School of Management, Walailak University, Nakhon Si Thammarat 80161, Thailand; tpavit@gmail.com; 3Food Biotechnology Research Team, Functional Ingredients and Food Innovation Research Group, National Center for Genetic Engineering and Biotechnology (BIOTEC), 113 Thailand Science Park, Pathum Thani 12120, Thailand; atikorn.pan@biotec.or.th (A.P.); natthaporn.pho@biotec.or.th (N.P.); at.kittipat@hotmail.com (K.A.); 4Department of Food Technology, Faculty of Applied Biological Sciences, Vinh Long University of Technology Education, Vinh Long 890000, Vietnam; quanth@vlute.edu.vn

**Keywords:** hybrid chicken, textural property, composition, color, meat quality

## Abstract

The quality characterization of different parts of male and female Ligor hybrid chickens was investigated and compared with those of commercial broiler. Genotypes, muscle types, and sex had effects on the composition, physicochemical, and textural properties of chicken samples. Ligor hybrid chicken contained higher percentages of protein, moisture, ash, and collagen content but lower fat content than those of commercial broiler (*p* < 0.05), except in the case of breast, where no significant difference in moisture and ash was observed (*p* ≥ 0.05). The pH in breast meat of both chickens was lower than that of thigh meat. The color (*L**, *a**, and *b**) values of male and female chickens were not significantly different, except for the *L** value of broiler chicken, which was higher in female chickens than in male chickens. Higher cooking loss and shear force were found in male Ligor hybrid chicken. A similar protein pattern was observed for the protein from the same muscle type, irrespective of sex and genotype tested. It was observed that Ligor hybrid chicken contained higher glutamic acid and aspartic acid than commercial broilers. Therefore, Ligor hybrid chicken is a promising new source of nutrition, which can be beneficial for consumers.

## 1. Introduction

Nowadays, consumers with increasing health consciousness are becoming more aware of the nutritional value of the food they eat. This knowledge has led to an increase in the emphasis on food labels such as light, lean, low-fat, reduced calories, etc. [1]. One of the good sources of protein with a lower fat and cholesterol content than red meat is chicken meat. It is also affordable, typically with convenient portions and no religious prohibitions on its consumption [2]. Broiler chicken meat is considered commercial chicken meat. Mass production of broilers has already been achieved. A fast-growing broiler can fully grow within 5 to 6 weeks [3]. However, a higher fat content in the breast and thigh of the broiler when compared with indigenous chicken has been reported [4]. This could persuade customers who are concerned about their health to seek out poultry as a source of protein. Indigenous chicken, which has a unique taste with a chewy texture and high human-health value, is regarded as a delicacy, and is popular among consumers, particularly in Asian countries. It normally takes about 14 to 23 weeks to reach an optimum size for the market [5]. The demand for indigenous chickens is always increasing, although only a small portion of this demand can be met. This is the consequence of low production volume because of its slow growth rate, non-uniform size, and irregular quality, resulting in high production costs [6]. To tackle such a drawback, various crossbreed chickens have been developed in many Asian countries such as China [7,8], India [9], Pakistan [10], Bangladesh [11], Korea [12], and Thailand [3,13].

Ligor hybrid chicken is a new crossbreed between Dang Suratthani sires (Thai indigenous chickens) and Suranaree University of Technology (SUT) 101 chicken dams (a crossbreed between layer chickens and broiler), which has exhibited better growth performance than its sires (indigenous chickens) [6]. Since consumers judge meat quality from its appearance, texture, juiciness, firmness, tenderness, water holding capacity (WHC), odor, and flavor [14], according to Cross et al. [15], those features are among the most important and perceptible ones that influence the judgment of quality by consumers. Furthermore, the quality of poultry meat is influenced by quantifiable properties of meat such as WHC, shear force, drip loss, cooking loss, pH, and collagen content, all of which are indispensable for processors involved in the manufacture of value-added meat products [14]. Several factors such as genotype (breed and strain), race, type of muscle, sex, genetic origin, and slaughtering age can affect poultry meat quality [2]. The factor that has the most influence is the genotype of poultry. Debut et al. [16] showed significant differences in most of the meat quality indicators between a slow-growing French Label-type line and a fast-growing standard line of chickens exposed to different pre-slaughter stress conditions when estimating breast and thigh meat quality (pH decline, color, drip loss, and curing-cooking yield). They found that the breast muscle of Label chickens had higher reserves of glycogen at the time of slaughtering than the standard chickens owing to the variability in muscle fiber structure among different genetic types. In fact, a study by Lonergan et al. [17] comparing the meat quality of five genetic groups of chicken, namely inbred Leghorn, inbred Fayoumi, commercial broilers, F5 broiler-inbred Leghorn cross, and F5 broiler-inbred Fayoumi cross showed high differences in breast meat composition and quality. The other important factors were the type of muscle and sex. The influence of the type of muscle on the quality of chicken meat has been reported by several previous studies [2,3,4,10,13,17], while the effects of sex on carcass quality of various chicken strains were also studied by Reddy et al. [9], Shahin and Elazeem [18], and Kumar et al. [19]. However, the quality of Ligor hybrid chicken in terms of chemical composition and textural characteristics which are influenced by the type of muscle and sex has not been documented. Therefore, the aim of this study was to investigate the quality characteristics of breast, thigh, and skin of Ligor hybrid chickens from both sexes and compare them with commercial broiler.

## 2. Materials and Methods

### 2.1. Sample Preparation

Carcasses of male and female Ligor hybrid chickens, aged 14 weeks, live weights (1.5 ± 0.2 kg) were purchased from Smart Farm Walailak University, Nakhon Si Thammarat, Thailand. Commercial broilers, aged 6 weeks, of similar live weights (1.5 ± 0.2 kg) were purchased from Farmesh Southern Co., Ltd., Nakhon Si Thammarat, Thailand. The carcasses were immediately packed in a polystyrene box filled with ice and transported to the laboratory within 1 h. Upon arrival, the breast, thigh, and skin were separated from the carcasses and maintained at 4 °C. The color and pH of all samples were measured at 24 h postmortem. Shear force and cooking loss were also monitored within 48 h. For chemical analyses, the three different composite samples from different batches were used in this study. To maintain the sample for analysis, within 2 h after slaughtering, each sample was immediately minced using a food processor (Tefal, DO821838, Bangkok, Thailand), vacuum-packed, and stored at −20 °C until used for chemical analyses.

All experiments were performed with the approval of the Animal Ethics Committees of Walailak University (Protocol number: WU-ACUC-65006).

### 2.2. Determination of Proximate Composition

The proximate composition including moisture (A.O.A.C method number 950.46), crude protein (A.O.A.C method number 928.08, Kjeldahl factor of 6.25), fat (A.O.A.C method number 963.15), and ash (A.O.A.C method number 920.153) of all samples was analyzed [20].

### 2.3. Determination of Collagen Content

Hydroxyproline content was determined using Ehrlich’s reagent. A hydroxyproline standard curve was constructed using hydroxyproline ranging from 10 to 60 mg/L. Hydroxyproline content was calculated and converted to collagen content using the factor 7.25. The collagen content was reported as milligrams of collagen per gram of sample [4,21].

### 2.4. Determination of pH and Color

The pH of whole chicken samples was determined at 24 h postmortem by homogenizing the samples with distilled water at a ratio of 1:5 of (wt/vol). Then, the homogenates were measured using a pH meter (EUTECH, PH700, Singapore) [4,22]. For color measurement, minced samples were measured with a Hunter lab colorimeter (Color Flex, Hunter Lab Inc., Reston, VA, USA) consisting of a D65 illuminant and 10° observation angle, where the *L** (lightness), *a** (redness/greenness) and *b** (blueness/yellowness) values were recorded [23].

### 2.5. Determination of Shear Force and Cooking Loss

Shear force was determined on raw and cooked muscle samples (breast and thigh). To prepare cooked muscles, small pieces (1.5 × 3.0 × 0.5 cm) of each muscle from each chicken were cut, put in a tightly sealed plastic bag, and cooked in a water bath at 80 °C for 10 min. After being cooked, the samples were cooled to room temperature by exposure to cool running water. The muscle samples were removed from the plastic bag, blotted with filter paper, and weighed. The cooking loss was reported as a percentage of initial weight (wt/wt, wet basis). Muscle samples, raw and cooked, were then cut to 1.0 × 2.0 × 0.5 cm for shear analysis using a texture analyzer (Model TA-XT2, Stable MicroSystems, Surrey, UK) with a load cell of 50 kg equipped with a Warner-Bratzler shear apparatus [3,24]. The operating parameters consisted of a cross-head speed of 2 mm/s. The shear force perpendicular to the axis of muscle fibers was recorded as the shear force value.

### 2.6. SDS-Polyacrylamide Gel Electrophoresis (SDS-PAGE)

Protein patterns were analyzed by SDS-PAGE according to the method of Laemmli [25]. The samples were mixed with sample buffer (50 mM Tris-HCl, pH 6.8, containing 4% SDS, 20% glycerol, 10% β-mercaptoethanol). The samples (10 μg protein) were loaded onto the polyacrylamide gel made of 10% running gel and 4% stacking gel and subjected to electrophoresis at a constant current of 15 mA per gel using a Mini Protein II unit (Bio-Rad Laboratories Inc., Richmond, CA, USA). After separation, the proteins were stained with 0.02% (*w*/*v*) Coomassie Brilliant Blue R-250 in 50% (*v*/*v*) methanol, 7.5% (*v*/*v*) acetic acid, destained with 50% methanol (*v*/*v*) and 7.5% (*v*/*v*) acetic acid, followed by 5% methanol (*v*/*v*) and 7.5% [26].

### 2.7. Amino Acid Profile

The chicken sample was hydrolyzed using 6N HCl for 18 h at 110 °C prior to amino acid analysis. After finishing the hydrolyzation process, the hydrolysate was filtered using a 0.22 µm nylon membrane syringe filter. Then, 25 µL of filtered hydrolysate was subjected to amino acid determination by HPAEC-IPAD. The high-performance anion-exchange chromatography coupled with integrated pulsed amperometric detection (Thermo Fisher Scientific, Waltham, MA, USA) was used to quantify amino acid in chicken hydrolysate samples. For amino acid detection, a detection waveform “Gold, pH/Ag/AgCl RE, AAA” was used and the reference electrode was pH mode. The chromatographic separation was performed on a 2 × 250 mm AminoPac PA10 Analytical column (Thermofisher) connected with a 2 × 50 mm AminoPac PA10 Guard column (Thermofisher). The column and compartment temperature were both maintained at 30 °C and 25 °C, respectively. The operated gradients were 18.2 megaohm-cm water (solvent A), 250 mM Sodium hydroxide in water (solvent B), and 1 M sodium acetate in water (Solvent C) with a flow rate of 0.25 mL/min. Amino acid standards were used to generate a standard curve to quantify the content of each targeted compound in this study.

### 2.8. Statistical Analysis

A pairwise *T*-test was applied to compare the difference between genotypes (Ligor hybrid chicken and broiler). The difference between sex (male and female) was analyzed using a pairwise *T*-test. The effect of muscle types of chicken (breast, thigh, and skin) in each measured parameter was analyzed using variance analysis (ANOVA). The contrast of means was rendered by the multiple range test of the Duncan. Using SPSS for Windows (SPSS Inc., Chicago, IL, USA), statistical analysis was performed. Data with *p* < 0.05 were considered to be statistically significant.

## 3. Results and Discussion

### 3.1. Proximate Composition

The proximate composition of breast, thigh, and skin of broiler and Ligor hybrid chicken from both sexes are presented in Table 1. Ligor hybrid chicken contained higher percentages of protein, moisture, and ash but lower fat content than commercial broiler (*p* < 0.05), except in the case of breast, no significant difference in moisture and ash was observed (*p* ≥ 0.05). The lower fat content in Ligor hybrid chicken might be caused by the lipid metabolism in indigenous and/or hybrid chickens occurring to a greater extent than that in commercial broiler chickens. According to Zheng et al. [27], indigenous chicken showed higher levels of liver proteins involved in lipid degradation. This result could reconfirm that higher lipid metabolism could, therefore, be one of the reasons for the lower fat content in Ligor hybrid chicken. A similar result was observed in Korat hybrid chicken [3], 4-lines, and 5-lines cross Thai hybrid native chicken [28]. Chuaynukool et al. [29] also found that genotypes of chicken play an important role in carcass fatness. The skin showed the highest fat content followed by thigh and breast, respectively, while comparing the same genotype and sex (*p* < 0.05). It is known that chicken breast accumulates glycogen as a source of energy while the thigh and skins store fat [30,31]. These results are in agreement with those reported by Tan et al. [32]. They found that the chicken skin has a higher fat content as compared to chicken meat [32]. Considering the same meat types with the same genotype, the female possessed a higher fat content for both thigh and breast (*p* < 0.05 by *T*-test). These results were in accordance with those of a previous study by Reddy et al. [9] who reported that the fat percent of female Rajasri chicken was higher than males in both thigh and breast muscles. For protein content, all chicken parts of both Ligor hybrid chickens contained higher amounts of protein when compared with those of commercial broiler (*p* < 0.05). The highest protein content (23.81%) was found in the breast of Ligor hybrid chickens (*p* < 0.05). However, there were no differences in protein content between males and females, regardless of genotypes and muscle types (*p ≥* 0.05 by *T*-test). The higher protein content in Ligor hybrid chicken could be due to the effect of growth rate and age on the protein content of poultry, in which protein content of poultry increased with an increase in animal age [33]. In the present study, slow-growing Ligor hybrid chickens (aged 14 weeks) were older than a fast-growing broiler chicken (aged 6 weeks) of the same weight. Katemala et al. [3] found that older Korat hybrid chicken had a higher protein content than younger commercial broiler. Overall, the proximate composition of Ligor hybrid chicken was significantly different from that of broiler, which was affected by genotype, muscle types, and sex, particularly in protein and fat contents. This factor may influence the physicochemical and textural properties of chicken, which in turn affects chicken quality.

### 3.2. Collagen Content

The collagen content of male and female Ligor hybrid chickens for different muscle types, breast, thigh, and skin, in comparison with those of commercial broiler, is shown in Figure 1. There was a similar profile of collagen content in different muscle types for commercial broiler and Ligor hybrid chickens. The skin showed the highest collagen content followed by thigh and breast, respectively, while comparing chicken of the same sex (*p* < 0.05). It is well known that the skin of an animal that has a tough and strong matrix to protect the body, is a good source of collagen [21,34]. For breast meat, there was no significant difference in collagen content, regardless of sex and genetic types (*p ≥* 0.05). On the other hand, more variation was observed from thigh and skin. Considering the same meat types, the male possessed a higher collagen content for both thigh and skin (*p* < 0.05). In addition, in the same sex, the commercial broiler revealed a lower collagen content than the Ligor hybrid (*p* < 0.05). These results were in accordance with those of a previous study by Tougan et al. [14] and Jaturasitha et al. [2]. They found that the quantity of collagen could determine the tenderness of the meat, where the thigh muscle had a higher collagen content than the breast. The decrease in tenderness in chicken meat could be due to the increase in collagen content [14,35]. Comparing thigh and breast muscles, a higher variation in collagen content in thigh muscle among genetic types could probably result in a large difference in meat texture. Additionally, smaller diversity in the collagen of breast muscle from different genotypes may correspond to a smaller difference in texture [2,36]. Collagen content is one of the quantifiable properties of meat, in which sex, maturity, species, and muscle type of poultry are the crucial factors directly governing the toughness of animal meat [28,37]. Given that the indigenous Dang Surathani sires used for developing the Ligor hybrid chicken were used for cock fighting, this is suspected to be a reason for their having a high collagen muscle content with tough meat when compared with the tender meat of broilers [2,38]. The effect of sex on the meat quality of the chicken was reported by Hussein et al. [39]. Sex could influence the myofibrillar fragmentation index (MFI) and tenderness, in which females could give a higher MFI with more tender muscle than males. This could relate to the difference in metabolism and activity between male and female chickens [39,40]. Therefore, these results suggested that differences in sex and meat sections of Ligor hybrid chicken had an impact on collagen content, which was different from those of commercial broiler, particularly in thigh and skin sections. These could determine the toughness and meat properties of this new crossbreed chicken from Thailand.

### 3.3. Physicochemical Properties

The pH and color (*L*, a*,* and *b**) values of broiler and Ligor hybrid chickens are presented in Table 2. The pH of broiler chicken of all chicken parts was higher than Ligor hybrid chicken (*p* < 0.05). Previous studies also reported that breast and thigh meats of broiler chicken had higher pH than breast and thigh meats of Thai indigenous chicken [4] and hybrid chicken [3]. Katemala et al. [3] found that the pH values of breast and thigh meats decreased with the increasing age of Korat hybrid chicken. In the current study, slow-growing Ligor hybrid chickens (aged 14 weeks) were older than a fast-growing broiler chickens (aged 6 weeks). Diaz et al. [41] reported that older animals had a lower pH of meat than younger animals due to a higher content of muscle glycogen. The glycogen content in muscle is predominantly affected by the proportional changes in muscle fibers where the patterns of muscle metabolism may differ. For instance, the breast and thigh of older birds tended to have increased glycogen storage, thereby reducing the postmortem pH. The pH in breast meat of both chickens was lower than in thigh meat, which is in accordance with the finding of Katemala et al. [3]. Usually, breast meat consisted of type IIB fibers which had high glycogen content. This characteristic of breast meat is related with higher lactic acid accumulation postmortem than thigh meat [42].

The Ligor hybrid chicken had a significantly lower *L** value than broiler chicken (*p* < 0.05), except for the breast, which showed no significant differences in the *L** value. No significant differences in the *a** value of breast and skin were observed between broiler and Ligor hybrid chickens (*p* ≥ 0.05), whereas the *a** value of Ligor hybrid chicken thigh was significantly higher than that of broiler chicken thigh (*p* < 0.05). Furthermore, Ligor hybrid chicken showed a higher b* value than broiler chicken in all chicken parts and sex (*p* < 0.05). These results indicated that Ligor hybrid chicken had more yellow in breast, dark red in thigh, and dark yellow in skin compared with broiler chicken, probably due to the characteristics of Dang Suratthani sires (indigenous chicken). Wattanachant et al. [4] found that the indigenous chicken muscles had more red and yellow than broiler chicken. The poultry meat color was influenced by meat pH, myoglobin content, and the redox state of the myoglobin. Myoglobin content is related to species, animal age, and muscle type [43]. Moreover, the results also revealed that the Ligor hybrid chicken was more yellowish in skin color than broiler chicken. Fletcher [44] stated that the pigmentation in the skin of chicken depended on various factors including the dietary source of pigments, the genetic capability, chicken health, and processing. The genetic disposition of chicken could deposit the carotenoid pigment from the diet in the skin. The intensity of the yellow color in chicken is related to carotenoids (primarily xanthophylls in diet) [45].

As for the color of chicken parts, both chickens showed similar trends in color. The highest *L** and *b** values were observed in the skin, followed by breast and thigh meats, respectively (*p* < 0.05). In contrast, thigh meat had the highest *a** value compared with breast meat and skin (*p* < 0.05). Moreover, the higher *a** value and the lower *L** and *b** values were observed in the thigh meat compared with the breast meat of both chickens (*p* < 0.05). The dark red color of thigh meat was probably due to the predominance of myoglobin. Fleming et al. [46] found that the chicken thigh had a higher *a** value and lower *b** value compared with the breast of broiler chicken because the higher myoglobin content was observed in the chicken thigh. Moreover, a higher myoglobin content was noticed in thigh meat than that in breast meat of indigenous chicken [43]. The color of chicken skin is related to the carotenoids [45] and also fat content. Like the chemical composition of skin (Table 1), the fat content in the skin of both chickens was higher compared with chicken breast and thigh. Peña-Saldarriaga et al. [31] stated that the fatty deposits presented a yellow color when the chicken diet consisted of high carotenoids and xanthophylls. Moreover, fat color was also affected by the concentration of hemoglobin retained in the adipose tissue and also connective tissue [31]. The color (*L**, *a*,* and *b**) values of male and female chickens were not significantly different (*p* ≥ 0.05 by *T*-test), except for the *L** value of broiler chicken which was higher in females than that of male chickens (*p* < 0.05 by *T*-test).

### 3.4. Cooking Loss

Cooking loss of male and female Ligor hybrid chickens for different muscle types is shown in Figure 2, in comparison with those of commercial broiler. The muscles from Ligor hybrid chicken showed a higher cooking loss than those of commercial broiler compared to the same muscle type and sex (*p* < 0.05). This result was related to the collagen content (Figure 1). A similar result was reported by Jaturasitha et al. [47] who studied meat characteristics of male chickens between Thai indigenous compared with improved layer breeds and their crossbred. They found that cooking losses of all muscle types (breast and thigh) were significantly different among genotypes. For both genotypes, the thigh muscle possessed a lower cooking loss than the breast muscle, regardless of sex (*p* < 0.05). The thigh of both male broiler and Ligor hybrid chickens had higher cooking loss than female (*p* < 0.05 by *T*-test). In contrast, there was no difference in cooking loss of breast muscle for males and females in both chickens (*p* ≥ 0.05 by *T*-test). The higher collagen content was found in muscles from Ligor hybrid chicken in comparison with broiler chicken (Figure 1). The collagen content of Ligor hybrid chickens is related to slow-growing (aged 14 weeks). Moreover, the previous study reported that collagen cross-linking of poultry meat increased with age [44]. Katemala et al. [3] found that insoluble collagen was found in Korat hybrid chicken higher than in broiler chicken. Wattanachant et al. [4] also found that the indigenous chicken muscles (breast and thigh muscles) showed higher cooking loss than broiler chicken due to the difference in total, insoluble, and soluble collagen contents of each muscle type. The lower collagen cross-linking in younger broiler chickens was disrupted by heating, whereas the higher collagen cross-linking remained insoluble and shrink in older indigenous chickens. Thus, this led to the loss of moisture due to the squeezing of the heat-denatured myofibrillar protein in meat muscles [4].

### 3.5. Textural Properties

The shear force of breast and thigh muscles of Ligor hybrid chicken from different sex as raw and cooked meat in comparison with those of commercial broiler is shown in Table 3. For raw meat, the muscles from Ligor hybrid chicken showed a higher shear force than those of commercial broiler compared to the same muscle type and sex (*p* < 0.05). For both genotypes, the thigh muscle possessed a higher shear force than the breast muscle, regardless of sex (*p* < 0.05). This result was in accordance with the collagen content (Figure 1). The higher collagen content of thigh muscle could render the tougher and chewier texture of chicken meat, in which the meat of females was less exudative and more tender than that of males [2,48]. Jaturasitha et al. [2] found that Thai indigenous chickens had a higher shear force and collagen content, particularly in thigh muscle than the meat of broilers. However, there was no difference in the collagen content of breast muscle for all samples tested (Figure 1) (*p* < 0.05). This might indicate the impact of other quantifiable properties of meat apart from collagen content, in which WHC, muscle fiber diameter, and fat content could also determine the qualities of meat [3,49]. It was noted that the shear values were increased after cooking for all muscles tested, while the meat from Ligor hybrid chicken showed more effect than the commercial broiler. This might relate to the heat denaturation of highly cross-linked collagen, especially in Ligor hybrid chicken. The insoluble and shrink collagen rendered the leaching out of the water from the heat-denatured myofibrillar protein, resulting in a tougher texture after cooking. Nevertheless, there was no significant difference in shear force for raw and cooked breast muscle from commercial broiler (*p* < 0.05), which had similar collagen content to those of Ligor hybrid chicken (Figure 1). This could be explained by a higher soluble collagen content of the younger chicken (6-week age), which was melted by heat [4,14]. The loss of WHC of muscle fiber due to heat denaturation could show more impact on changing of muscle texture property, which is reconfirmed by a higher cooking loss of Ligor hybrid breast muscle (Figure 2). A similar result was also reported for the texture of raw and cooked Thai indigenous and broiler chicken muscles [4].

### 3.6. Protein Patterns

The molecular weight profile of protein from different muscle types, breast (B), thigh (T), skin (S), female (F), and male (M) Ligor hybrid chickens in comparison with those of commercial broiler is presented in Figure 3. A similar protein pattern was observed for the protein from the same muscle type, irrespective of sex and genotype. Breast and thigh muscle contained myosin heavy chain (200 kDa), actinin (114 kDa), and actin (40 kDa), as the major myofibrillar component, where tropomyosin (37 kDa), light myosin (29 kDa), troponin (27 kDa), and myoglobin (16.5 kDa) were found as minor content. This result was in accordance with the protein pattern of chicken meat reported in the previous works [50,51]. Considering the same genotype, the breast muscle with a higher band intensity of those myofibrillar components was found in females of the Ligor hybrid but in the males of the broiler. The contrast result was obtained for thigh muscle. Female broiler thighs had the higher myofibrillar protein band intensity, while female Ligor hybrid thighs provided less, compared with their relative male. The results indicated the differences in the amount of muscle protein contents directly related to the difference in chicken sex and genotypes. In addition, the degradation and muscle fiber changing during the postmortem period could also differentiate based on their genotype [52]. In contrast, the skin section possessed a different protein pattern, compared with those from breast and thigh muscles, regardless of sex and genotype tested. The main protein bands were found at 205, 156, and 136 kDa, which related to β-chains, α1-chain, and α2-chain, the major components of skin collagen [21,34]. Notably, all skin samples tested showed a similar molecular weight profile and band intensity, which suggested the less impact of sex and genotype on the protein pattern of this section. Overall, the diverse protein pattern of chicken muscle could be influenced by muscle type, sex, and genotypes, in which the differences in protein components between Ligor hybrid and broiler chickens would govern their different characteristics and nutritional contents.

### 3.7. Amino Acid Composition

The amino acid composition of breast, thigh, and skin of broiler and Ligor hybrid chicken from different sexes is presented in Table 4. Of the 20 amino acids normally present in animal proteins, 10 can be synthesized in the cells; the other 10 amino acids either cannot be synthesized or are synthesized in quantities too small to supply the needs of the body. The latter amino acids are called essential amino acids because they must be supplied in the diet [53]. Essential and non-essential amino acids were found in both broiler and Ligor hybrid chicken. All chicken samples contained very high lysine, leucine, arginine, alanine, glutamic acid, and aspartic acid. It was noted that Ligor hybrid chicken had higher glutamic acid and aspartic acid than those commercial broilers (*p* < 0.05), regardless of muscle types and sex. Considering the same genotype and sex, both breast and thigh had higher glutamic acid and aspartic acid than skin (*p* < 0.05). Petcharat et al. [54] suggested that glutamic acid or aspartic acid are known to provide an umami taste. It could increase the overall taste of food by cooperating with other substances. Farmer [55] also found that glutamic acid played an essential role in the taste and flavor of chicken meat. A higher glutamic acid was also reported when compared between indigenous chicken and commercial broilers [4]. Moreover, the skin of broiler and Ligor hybrid chicken had higher content in glycine than breast and thigh. When comparing the same genotype and chicken parts, male chickens contained higher glycine than female chickens (*p* < 0.05 by *T*-test). This might be caused by the influence of collagen content in the samples. In general, glycine is found in the largest quantities of collagen, which is located at every third position of the α-chain (Gly-X-Y), except in the telopeptide regions [21]. These were reconfirmed with collagen content in Figure 1. Among all samples, the highest total amino acid was found in the breast of males in Ligor hybrid chicken (*p* < 0.05). The total amino acid of all samples was related to protein content (Table 1). Chicken samples with high protein content also had high content in total amino acids. Therefore, the results suggested that Ligor hybrid chicken could be a good choice of alternative protein source, as it contained amounts of all the amino acids essential in human nutrition and umami taste.

## 4. Conclusions

The genotype, muscle type, and sex were most likely responsible for the differences in composition, physicochemical, and textural aspects of broiler and Ligor hybrid chickens. Ligor hybrid chicken contained higher protein and collagen with lower fat when compared to commercial broiler. Faced with high cooking loss and shear force, especially in males, Ligor hybrid chicken might have specific customer segments. Furthermore, all essential amino acids, glutamic acid, and aspartic acid that provide an umami taste were also identified. Therefore, the new crossbreed Ligor hybrid chicken could be used as an alternative protein source with high nutritive values.

## Figures and Tables

**Figure 1 foods-11-01929-f001:**
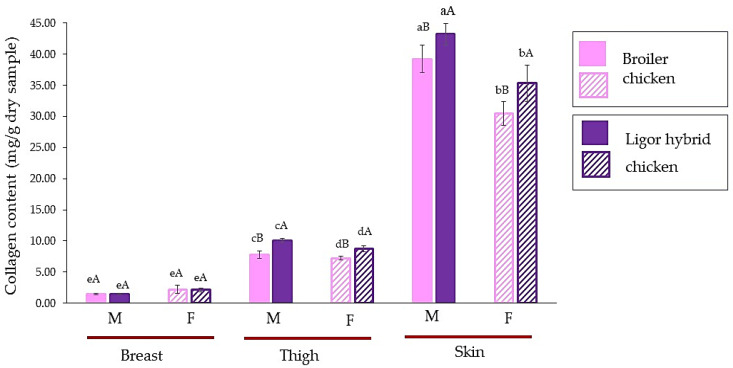
Collagen content (mg/g dry sample) of breast, thigh, and skin of broiler and Ligor hybrid chicken from different sexes. Different lowercase superscripts (a–e) on the bar within the same genotype of chicken indicate significant differences (*p* < 0.05). Different uppercase superscripts (A,B) on the bar under the same part and sex of chicken indicate significant differences (*p* < 0.05) by *T*-test. M, male chicken; F, female chicken.

**Figure 2 foods-11-01929-f002:**
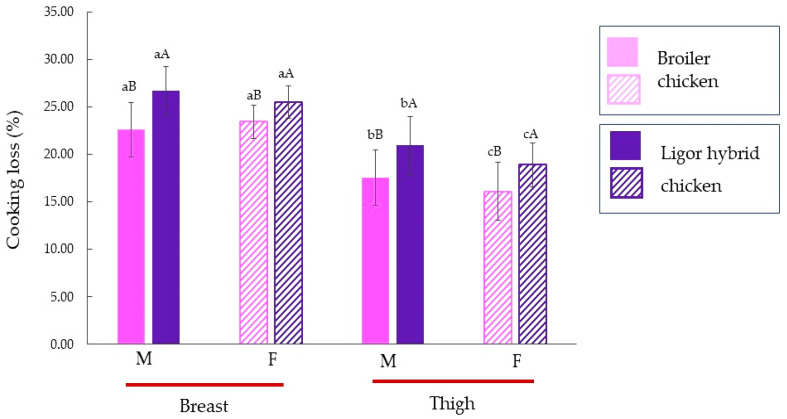
Cooking loss (%) of breast, thigh, and skin of broiler and Ligor hybrid chicken from different sex (*n* = 15). Different lowercase superscripts (a–c) on the bar within the same genotype of chicken indicate significant differences (*p* < 0.05). Different uppercase superscripts (A,B) on the bar under the same muscle type and sex of chicken indicate significant differences (*p* < 0.05) by *T*-test. M, male chicken; F, female chicken.

**Figure 3 foods-11-01929-f003:**
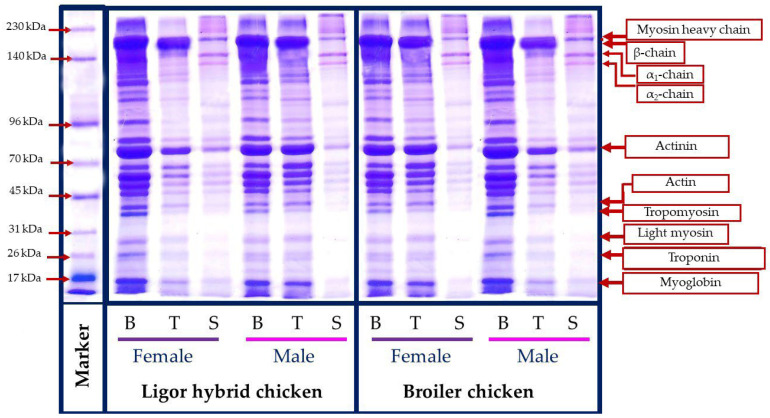
Protein patterns of breast, thigh, and skin of broiler and Ligor hybrid chicken from different sex. B, breast; T, thigh; S, skin female.

**Table 1 foods-11-01929-t001:** Proximate composition (%, wet basis) of breast, thigh, and skin of broiler and Ligor hybrid chicken from different sex.

Chicken	Chicken Parts	Sex	Moisture	Crude Protein	Fat	Ash
Broiler	Breast	M	73.84 ± 0.20 ^bA^	22.63 ± 0.83 ^aB^	0.34 ± 0.03 ^eA^	1.17 ± 0.06 ^aA^
		F	73.84 ± 0.34 ^bA^	21.80 ± 0.50 ^aB^	0.44 ± 0.01 ^dA^	1.19 ± 0.04 ^aA^
	Thigh	M	74.94 ± 0.02 ^aB^	19.21 ± 0.04 ^bB^	4.65 ± 0.12 ^cA^	0.88 ± 0.04 ^bB^
		F	74.89 ± 0.07 ^aB^	18.82 ± 0.55 ^bB^	5.32 ± 0.47 ^bA^	0.86 ± 0.06 ^bB^
	Skin	M	54.22 ± 0.17 ^cB^	9.69 ± 0.39 ^cB^	12.15 ± 0.42 ^aA^	0.37 ± 0.02 ^cB^
		F	53.96 ± 0.78 ^cB^	9.52 ± 0.77 ^cB^	12.70 ± 0.45 ^aA^	0.44 ± 0.04 ^cB^
Ligor	Breast	M	73.88 ± 0.34 ^bA^	23.81 ± 0.19 ^aA^	0.24 ± 0.03 ^fB^	1.15 ± 0.10 ^aA^
		F	73.66 ± 0.45 ^bA^	23.69 ± 0.09 ^aA^	0.33 ± 0.02 ^eB^	1.11 ± 0.06 ^aA^
	Thigh	M	76.72 ± 0.13 ^aA^	21.17 ± 0.57 ^bA^	0.65 ± 0.04 ^dB^	0.97 ± 0.02 ^abA^
		F	76.47 ± 0.16 ^aA^	21.11 ± 0.01 ^bA^	0.94 ± 0.02 ^cB^	0.96 ± 0.07 ^bA^
	Skin	M	60.46 ± 1.31 ^cA^	14.71 ± 0.50 ^cdA^	8.78 ± 0.17 ^bB^	0.52 ± 0.01 ^dA^
		F	59.05 ± 0.10 ^dA^	14.22 ± 0.27 ^dA^	9.22 ± 0.15 ^aB^	0.60 ± 0.02 ^cA^

Values are presented as mean ± SD (*n* = 3). Different lowercase superscripts (a–f) in the same column within the same genotype of chicken indicate significant differences (*p* < 0.05). Different uppercase superscripts (A,B) in the same column under the same part and sex of chicken indicate significant differences (*p* < 0.05) by *T*-test. M, male chicken; F, female chicken.

**Table 2 foods-11-01929-t002:** Physicochemical properties of breast, thigh, and skin of broiler and Ligor hybrid chicken from different sex.

Chicken	Chicken Parts	Sex	pH	*L**	*a**	*b**
Broiler	Breast	M	5.87 ± 0.01 ^dA^	60.78 ± 0.35 ^bA^	6.60 ± 0.42 ^bA^	15.60 ± 0.24 ^bB^
		F	5.84 ± 0.01 ^eA^	61.01 ± 0.61 ^bA^	6.81 ± 0.28 ^bA^	15.48 ± 0.39 ^bB^
	Thigh	M	6.44 ± 0.01 ^aA^	55.78 ± 1.07 ^dA^	8.55 ± 0.58 ^aB^	14.63 ± 0.66 ^cB^
		F	6.36 ± 0.01 ^bA^	57.02 ± 0.59 ^cA^	8.62 ± 0.21 ^aB^	14.52 ± 0.12 ^cB^
	Skin	M	6.20 ± 0.03 ^cA^	76.60 ± 0.39 ^aA^	4.66 ± 0.41 ^cA^	16.43 ± 0.60 ^aB^
		F	6.18 ± 0.02 ^cA^	76.15 ± 0.13 ^aA^	4.29 ± 0.31 ^cA^	16.63 ± 0.46 ^aB^
Ligor	Breast	M	5.51 ± 0.00 ^cB^	61.29 ± 0.65 ^bA^	6.41 ± 0.20 ^bA^	19.95 ± 0.17 ^bA^
		F	5.47 ± 0.01 ^dB^	60.10 ± 0.36 ^bA^	6.51 ± 0.20 ^bA^	20.25 ± 0.20 ^bA^
	Thigh	M	5.88 ± 0.01 ^aB^	54.04 ± 0.81 ^cB^	9.14 ± 0.88 ^aA^	17.84 ± 0.46 ^cA^
		F	5.76 ± 0.01 ^bB^	53.40 ± 0.85 ^cB^	10.32 ± 0.33 ^aA^	17.95 ± 0.71 ^cA^
	Skin	M	5.71 ± 0.05 ^bB^	73.26 ± 0.32 ^aB^	4.44 ± 1.46 ^cA^	23.89 ± 0.33 ^aA^
		F	5.74 ± 0.07 ^bB^	73.64 ± 0.26 ^aB^	4.54 ± 0.26 ^cA^	25.20 ± 0.87 ^aA^

Values are presented as mean ± SD (*n* = 3). Different lowercase superscripts (a–e) in the same column within the same genotype of chicken indicate significant differences (*p* < 0.05). Different uppercase superscripts (A,B) in the same column under the same part and sex of chicken indicate significant differences (*p* < 0.05) by *T*-test. M, male chicken; F, female chicken. *L**, lightness; *a**, redness/greenness; *b**, blueness/yellowness.

**Table 3 foods-11-01929-t003:** Shear force (kg) of breast and thigh of broiler and Ligor hybrid chicken from different sex.

Chicken	Chicken Parts	Sex	Condition		Significance
			Raw	Cooked	
Broiler	Breast	M	1.23 ± 0.27 ^cA^	1.55 ± 0.36 ^cB^	NS
		F	1.08 ± 0.15 ^cA^	1.40 ± 0.24 ^cB^	NS
	Thigh	M	2.44 ± 0.57 ^aB^	3.17 ± 0.56 ^aB^	*
		F	2.17 ± 0.32 ^bB^	2.81 ± 0.38 ^bB^	*
Ligor	Breast	M	1.32 ± 0.53 ^cA^	4.08 ± 1.30 ^cA^	*
		F	1.19 ± 0.45 ^cA^	3.44 ± 0.99 ^dA^	*
	Thigh	M	3.28 ± 0.40 ^aA^	8.73 ± 1.42 ^aA^	*
		F	3.07 ± 0.21 ^bA^	6.25 ± 1.19 ^bA^	*

Values are presented as mean ± SD (*n* = 15). Different lowercase superscripts (a–d) in the same column within the same genotype of chicken indicate significant differences (*p* < 0.05). Different uppercase superscripts (A,B) in the same column under the same muscle type and sex of chicken indicate significant differences (*p* < 0.05) by *T*-test. * Significant differences (*p* < 0.05) between raw and cooked samples were determined by a *T*-test. NS, not significant; M, male chicken; F, female chicken.

**Table 4 foods-11-01929-t004:** Amino acid composition (g/100 g dry sample) of breast, thigh, and skin of broiler and Ligor hybrid chicken from different sex.

Sample	LF-S	LF-B	LF-T	LM-S	LM-B	LM-T	BF-S	BF-B	BF-T	BM-S	BM-B	BM-T
Essential Amino Acids												
Lys	2.18 ± 0.70 ^fA^	6.51 ± 0.41 ^cA^	5.92 ± 0.28 ^dA^	3.22 ± 0.71 ^eA^	8.68 ± 0.34 ^aA^	7.42 ± 0.28 ^bA^	0.92 ± 0.11 ^eB^	6.49 ± 0.13 ^bA^	5.46 ± 0.31 ^cB^	1.57 ± 0.48 ^dB^	8.38 ± 0.76 ^aA^	6.08 ± 1.72 ^bcB^
Thr	0.97 ± 0.24 ^cA^	2.04 ± 0.28 ^aB^	2.25 ± 0.38 ^aA^	1.31 ± 0.22 ^bA^	2.81 ± 0.53 ^aB^	2.99 ± 0.43 ^aA^	0.50 ± 0.09 ^dB^	2.59 ± 0.17 ^bA^	2.53 ± 0.15 ^bA^	0.74 ± 0.21 ^cB^	3.23 ± 0.18 ^aA^	2.67 ± 0.79 ^abB^
Val	0.63 ± 0.24 ^bA^	1.15 ± 0.15 ^aB^	1.24 ± 0.12 ^aB^	0.72 ± 0.14 ^bA^	1.33 ± 0.18 ^aB^	1.48 ± 0.16 ^aB^	0.32 ± 0.25 ^bB^	1.63 ± 0.26 ^aA^	1.48 ± 0.11 ^aA^	0.46 ± 0.15 ^bB^	1.71 ± 0.33 ^aA^	1.53 ± 0.36 ^aA^
Ile	1.20 ± 0.22 ^dA^	3.50 ± 0.50 ^bA^	3.24 ± 0.34 ^bA^	1.72 ± 0.26 ^cA^	4.80 ± 1.08 ^aA^	4.36 ± 0.85 ^aA^	0.46 ± 0.07 ^dB^	3.60 ± 0.40 ^bA^	2.90 ± 0.33 ^bA^	0.84 ± 0.24 ^cB^	5.03 ± 0.57 ^aA^	4.84 ± 0.48 ^aA^
Leu	2.79 ± 0.67 ^fA^	6.99 ± 0.29 ^cA^	6.47 ± 0.69 ^dA^	3.64 ± 0.57 ^eA^	9.85 ± 1.87 ^aA^	8.25 ± 1.19 ^bA^	1.18 ± 0.25 ^fB^	6.90 ± 0.23 ^cA^	5.87 ± 0.20 ^dA^	2.00 ± 0.51 ^eB^	9.10 ± 1.06 ^aA^	8.49 ± 1.99 ^bA^
Met	0.41 ± 0.12 ^eA^	1.23 ± 0.16 ^cB^	1.22 ± 0.16 ^cA^	0.50 ± 0.12 ^dA^	1.78 ± 0.41 ^aB^	1.49 ± 0.30 ^bA^	0.15 ± 0.04 ^fB^	1.41 ± 0.15 ^bA^	1.11 ± 0.21 ^dA^	0.21 ± 0.07 ^eB^	1.91 ± 0.35 ^aA^	1.26 ± 0.44 ^cB^
His	0.83 ± 0.20 ^fA^	2.26 ± 0.34 ^cA^	2.05 ± 0.32 ^dA^	1.16 ± 0.24 ^eA^	3.41 ± 0.70 ^aA^	2.69 ± 0.44 ^bA^	0.27 ± 0.08 ^fB^	2.35 ± 0.26 ^bA^	1.84 ± 0.21 ^dB^	0.43 ± 0.17 ^eB^	3.18 ± 0.22 ^aB^	2.00 ± 0.47 ^cB^
Phe	1.21 ± 0.34 ^fA^	3.00 ± 0.20 ^cA^	2.77 ± 0.18 ^dA^	1.72 ± 0.29 ^eA^	4.43 ± 0.98 ^aA^	3.63 ± 0.59 ^bA^	0.56 ± 0.09 ^fB^	3.01 ± 0.22 ^bA^	2.55 ± 0.18 ^dB^	0.76 ± 0.23 ^eB^	4.14 ± 0.54 ^aB^	2.76 ± 0.90 ^cB^
Trp	0.20 ± 0.02 ^fA^	0.59 ± 0.08 ^cB^	0.46 ± 0.05 ^dB^	0.25 ± 0.05 ^eA^	0.97 ± 0.26 ^aB^	0.82 ± 0.16 ^bA^	0.07 ± 0.02 ^fB^	0.68 ± 0.09 ^bA^	0.60 ± 0.08 ^bA^	0.14 ± 0.05 ^eB^	1.08 ± 0.18 ^aA^	0.59 ± 0.20 ^cB^
**Non-essential amino acids**												
Arg	3.39 ± 1.45 ^aA^	3.76 ± 1.29 ^aA^	6.19 ± 4.24 ^aA^	2.96 ± 0.19 ^aA^	2.32 ± 0.82 ^aA^	2.69 ± 0.10 ^aA^	3.08 ± 3.75 ^aA^	2.47 ± 0.97 ^aB^	2.71 ± 0.58 ^aB^	2.88 ± 0.87 ^aB^	2.11 ± 0.47 ^aA^	2.56 ± 0.89 ^aB^
Ala	5.05 ± 1.14 ^fA^	8.49 ± 1.30 ^bA^	6.63 ± 0.59 ^dA^	6.30 ± 0.76 ^eA^	9.89 ± 1.82 ^aA^	7.68 ± 1.38 ^cA^	1.51 ± 0.21 ^fB^	7.03 ± 0.50 ^bB^	5.24 ± 0.54 ^dB^	2.13 ± 0.62 ^eB^	7.96 ± 0.51 ^aB^	5.47 ± 1.26 ^cB^
Gly	2.21 ± 0.51 ^bA^	0.89 ± 0.18 ^fB^	1.26 ± 0.24 ^dB^	2.67 ± 0.22 ^aA^	1.05 ± 0.41 ^eA^	1.35 ± 0.36 ^cB^	1.73 ± 0.29 ^bB^	1.18 ± 0.19 ^eA^	1.45 ± 0.16 ^cA^	2.41 ± 0.81 ^aB^	1.27 ± 0.13 ^dA^	1.77 ± 0.35 ^bA^
Ser	0.63 ± 0.21 ^dA^	2.49 ± 0.13 ^bB^	2.51 ± 0.72 ^bA^	0.69 ± 0.18 ^cA^	3.97 ± 1.00 ^aA^	3.21 ± 0.61 ^bA^	0.23 ± 0.06 ^cB^	2.84 ± 0.24 ^bA^	2.55 ± 0.34 ^bA^	0.29 ± 0.07 ^cB^	3.81 ± 0.22 ^aA^	2.65 ± 0.82 ^bA^
Pro	0.64 ± 0.34 ^bA^	0.23 ± 0.19 ^cA^	0.27 ± 0.20 ^cA^	0.90 ± 0.18 ^aA^	0.23 ± 0.08 ^cA^	0.27 ± 0.08 ^cA^	0.34 ± 0.04 ^bB^	0.23 ± 0.02 ^cA^	0.21 ± 0.06 ^cB^	0.52 ± 0.29 ^aB^	0.17 ± 0.03 ^cB^	0.21 ± 0.08 ^cA^
Glu	18.43 ± 1.18 ^dA^	26.02 ± 1.14 ^bA^	22.51 ± 2.39 ^cA^	19.26 ± 1.97 ^dA^	30.40 ± 3.70 ^aA^	26.31 ± 2.18 ^bA^	11.55 ± 2.15 ^eB^	15.82 ± 2.32 ^dB^	16.60 ± 1.75 ^dB^	18.51 ± 1.67 ^cB^	25.17 ± 1.33 ^aB^	21.38 ± 2.73 ^bB^
Asp	1.44 ± 0.47 ^eA^	4.57 ± 0.42 ^cA^	3.99 ± 0.21 ^dA^	1.92 ± 0.41 ^dA^	5.88 ± 1.18 ^aA^	5.12 ± 0.80 ^bA^	0.38 ± 0.16 ^eB^	4.39 ± 0.66 ^bB^	3.44 ± 0.64 ^cB^	0.74 ± 0.33 ^dB^	5.18 ± 0.31 ^aB^	3.78 ± 1.16 ^cB^
Cys	0.84 ± 0.18 ^eA^	2.11 ± 0.30 ^cA^	1.82 ± 0.41 ^cA^	1.38 ± 0.33 ^dA^	3.21 ± 0.67 ^aA^	2.74 ± 0.58 ^bA^	0.41 ± 0.18 ^fB^	2.14 ± 0.23 ^bA^	1.78 ± 0.09 ^dB^	0.54 ± 0.19 ^eB^	3.10 ± 0.44 ^aA^	1.92 ± 0.66 ^cB^
Tyr	0.90 ± 0.18 ^fA^	2.62 ± 0.12 ^cA^	2.37 ± 0.16 ^dA^	1.25 ± 0.27 ^eA^	3.77 ± 0.87 ^aA^	3.17 ± 0.46 ^bA^	0.43 ± 0.07 ^fB^	2.71 ± 0.14 ^bB^	2.22 ± 0.18 ^dB^	0.58 ± 0.18 ^eB^	3.61 ± 0.51 ^aB^	2.40 ± 0.71 ^cB^
Total	43.93 ± 6.15 ^fA^	77.75 ± 3.40 ^cA^	73.16 ± 6.58 ^dA^	54.59 ± 6.20 ^eA^	98.78 ± 3.57 ^aA^	84.68 ± 6.86 ^bA^	24.09 ± 3.34 ^fB^	67.45 ± 1.85 ^cB^	60.54 ± 3.76 ^dB^	35.96 ± 6.68 ^eB^	95.35 ± 8.81 ^aB^	70.97 ± 5.52 ^bB^

Values are presented as mean ± SD (*n* = 3). Different lowercase superscripts (a–f) in the same row within the same type of chicken indicate significant differences (*p* < 0.05). Different uppercase superscripts (A,B) in the same row under the same part and sex of chicken indicate significant differences (*p* < 0.05) by *T*-test. LF-S, Ligor hybrid chicken-Female-Skin; LF-B, Ligor hybrid chicken-Female-Breast; LF-T, Ligor hybrid chicken-Female-Thigh; LM-S, Ligor hybrid chicken-Male-Skin; LM-B, Ligor hybrid chicken-Male-Breast; LM-T, Ligor hybrid chicken-Male-Thigh; BF-S, Broiler chicken-Female-Skin; BF-B, Broiler chicken-Female-Breast; BF-T, Broiler chicken-Female-Thigh; BM-S, Broiler chicken-Male-Skin; BM-B, Broiler chicken-Male-Breast; BM-T, Broiler chicken-Male-Thigh.

## Data Availability

Data is contained within the article.

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
