# Peer review of "Quality Characterization of Different Parts of Broiler and Ligor Hybrid Chickens"

_foods, 2022, doi:10.3390/foods11131929_

Round 1

Reviewer 1 Report

Comments to the Author

This study was conducted to investigate the quality characterization of breast, thigh and skin of commercial broiler and Ligor hybrid chicken from different sex. The experiment design and the aims are good, but there are some problems based on this manuscript:

Throughout manuscript: the quality of English language needs to be improved.

Abstract

L 19-20: “The quality characterization of breast, thigh and skin of broiler and Ligor hybrid chicken from different sex compare with commercial broiler was investigated”. This sentence is hard to understand, please re-written.

L 21: had affected on... should be could affect the... or had effects on...

Introduction

L 37: delete reduced-fat

L 88: ...Thailand and commercial broilers... should be Thailand. Commercial broilers...

Materials and Methods

L 87: the live weight of Ligor hybrid chickens was?

L 90: how chickens are slaughtered?

L 94-96: When was the meat collected and stored at -20°C for chemical analysis?

L 113-116: Need to know illuminant and observer angle used.

Results and Discussion

L 229: “There results... should be “These results...

L 233-237: Very long and confusing sentence. Revise please.

L 241: delete very 

L 264: Table 2

L 266: pH higher should be higher pH

L 273: thighs should be thigh

L 289: than those of

L 313-314: ...the skin of both chickens consisted of high fat content compared with...replace with ...the fat content in the skin of both chickens was higher compared with... 

Reviewer 2 Report

It is an interesting study and falls within the scope and interests of the journal. Major short comings in the manuscript are listed below:

1. Language of the manuscript is very poor even first word of ‘Introduction’ is incorrect. There are numerous language, grammar, punctuation, and structural mistakes in the write-up. Therefore, the manuscript requires rigorous editing and proofreading by a native English-speaking colleague or a language editing service.

2. I am a little skeptical of the statistical analysis. The selection of ANOVA is misleading since there were only two groups i.e. sex (in each chicken part within the groups) and chicken parts (between the groups regardless of the sex). Ideally, the authors should have applied t-test to ascertain the differences within and between the groups. It is understandable that t-test is ANOVA between two groups and that the results might not change, however, it is factually incorrect to apply ANOVA. I advise the authors to revise the statistical analysis, and results and discussion (provided the results change after applying the t-test).

3. Number of replicates/samples (n = 3) for proximate, collagen, pH, color, water holding capacity, cooking loss, and texture are too low for reliability of the results. The authors should consider adding more data/replicates to make the results more reliable.
